# Regional Inequities in the Distribution of the Nursing Workforce in Italy

**DOI:** 10.3390/nursrep15070235

**Published:** 2025-06-27

**Authors:** Ippolito Notarnicola, Blerina Duka, Eriola Grosha, Giovanni Gioiello, Maurizio Zega, Rosario Caruso, Sara Carrodano, Gennaro Rocco, Alessandro Stievano

**Affiliations:** 1Department Medicine and Surgical, University of Enna “Kore”, 94100 Enna, Italy; giovanni.gioiello@unikore.it; 2Faculty of Medicine and Surgical, Catholic University “Our Lady of Good Counsel”, 1000 Tirana, Albania; bleriduka@yahoo.it; 3Department of Medicine and Surgical, University of Rome “Tor Vergata”, 00133 Rome, Italy; e.grosha8400@stud.unizkm.al; 4Centre of Excellence for Nursing Scholarship, OPI, 00149 Rome, Italy; maurizio.zega@fnopi.it (M.Z.); genna.rocco@gmail.com (G.R.); 5Department of Biomedical Sciences for Health, University of Milan, 20122 Milan, Italy; rosario.caruso@unimi.it; 6Clinical Governance and Risk Management Department, ASST 5, 19100 La Spezia, Italy; carrodano.sara@gmail.com; 7Department of Clinical and Experimental Medicine, University of Messina, 98122 Messina, Italy; alessandro.stievano@unime.it

**Keywords:** Human Development Index, equity, nursing workforce, Gini coefficient, regional disparities

## Abstract

**Background/Objectives**: Inequalities in access to nursing professionals represent a significant challenge to achieving equity in healthcare systems. In decentralized countries like Italy, disparities in the distribution of nurses persist despite a universal national health system. This study investigates the extent and determinants of regional inequality in the distribution of the nursing workforce in Italy. **Methods**: A retrospective ecological analysis was conducted using administrative data from official national sources (ISTAT, Ministry of Health) concerning the number of nurses and population per region, along with Human Development Index (HDI) data from 2021. Descriptive statistics, the Gini coefficient, Lorenz curve, and Pearson correlation were used to assess inequality and identify influencing factors. **Results**: The national Gini coefficient was 0.136, indicating a moderate degree of inequality in the distribution of nurses across Italian regions. A strong positive correlation was observed between HDI and nurse-to-population ratio (r = 0.76, *p* < 0.001), suggesting that more developed regions have higher nursing density. **Conclusions**: Despite a universal healthcare system, Italy shows persistent regional disparities in nurse distribution. These findings emphasize the need for targeted policies and coordinated planning to reduce inequalities and ensure equitable access to nursing care across all regions.

## 1. Introduction

The equitable distribution of health resources is a crucial element in ensuring access to care services and promoting an inclusive health system, capable of responding effectively to the health needs of the population [1,2]. However, in many territorial realities, significant inequalities persist in the availability of health personnel and in the accessibility to essential services [3]. Italy, like many other nations, has significant disparities in the distribution of the nursing workforce between different regions, influenced by socioeconomic, structural and health development factors. Regions with advanced health infrastructure have more nurses, while rural or less developed areas face staff shortages [4]. This imbalance negatively impacts the effectiveness and efficiency of the health system, limiting the ability to provide timely and adequate care to the entire population and exacerbating pre-existing health inequalities [5].

Nursing staff shortages in some geographical areas compromise the quality of care and can lead to increased waiting times, reduced quality of care and worsening health outcomes, especially for people living in disadvantaged regions [6]. In addition, these shortages can lead to overload for existing nursing staff, exposing them to high levels of stress and burnout, with direct consequences on job satisfaction and the turnover rate of healthcare staff [7]. A fair and efficient health system therefore requires targeted strategies to reduce disparities in the distribution of nursing resources and improve accessibility to health services throughout the country [8,9].

In this context, the Gini coefficient represents a useful tool for quantifying and analyzing inequalities in the distribution of nursing staff among Italian regions. Traditionally used to measure wealth distribution, the Gini coefficient has also been successfully applied in healthcare to assess resource distribution and disparities in access to services [1]. A Gini value close to zero indicates an equal distribution of nurses among the regions, while higher values suggest a concentration of resources in some areas at the expense of others. Utilizing this indicator facilitates an objective assessment of regional disparities, thereby providing a robust foundation for health planning and policy formulation aimed at addressing existing inequalities.

Italy has a health system characterized by marked regional differences in the quality and availability of the services offered. Northern regions, which are generally more industrialized and have a higher Gross Domestic Product (GDP) pro capita, tend to benefit from a higher availability of health resources, while southern and island regions face challenges related to economic difficulties, infrastructure gaps and lower investment in the health sector [10,11,12]. These differences translate into a highly variable quality of nursing care between different regions, directly affecting the ability of the health system to respond equitably to the needs of the population [13,14].

Although this study focuses on the nursing workforce, it is important to acknowledge that disparities in the distribution of other healthcare professionals—such as physicians, physiotherapists, and diagnostic technicians—can also influence access to care and patient safety. Several studies in the international literature have highlighted how imbalances across different professional groups may contribute to system-wide inequalities and inefficiencies. These disparities are particularly relevant in decentralized health systems, where regional autonomy can result in uneven allocation of human resources. While this analysis centers on nursing, future research should also investigate the distribution of other categories of health professionals to provide a more comprehensive picture of equity in healthcare access.

The Gini coefficient quantifies regional inequalities and informs health resource planning [15]. The information deriving from its application could support policy makers in designing economic and professional incentives aimed at incentivizing the presence of nurses in underserved areas [16]. Strategies such as financial bonuses, tax breaks or continuing education programs could help make nursing work more attractive in regions with greater staff shortages [17,18]. In addition, a better distribution of nursing staff could foster greater equity in access to care and improve population health outcomes, reducing existing disparities [19,20].

The adoption of a fair distribution of nursing staff is a priority to ensure the effectiveness of the national health system [21]. In the context of an aging population and an increase in demand for healthcare, it is essential to develop targeted strategies to optimize the distribution of human resources and improve accessibility to services [22].

Although this study specifically focuses on the nursing workforce, it is important to acknowledge that regional disparities also affect other professional categories, including physicians, allied health professionals, and support staff. Several studies have shown that uneven distribution of physicians across territories, particularly between urban and rural or peripheral areas, can result in delayed access to diagnosis and treatment, negatively impacting patient safety and health outcomes [21,22]. Moreover, these inter-professional imbalances may compound existing inequities when shortages in one category are not offset by others. Recognizing the systemic nature of health workforce distribution is essential to designing integrated policies that address the needs of all healthcare professions. However, given the unique role of nurses in providing continuous and widespread care, particularly in primary and community settings, the present study concentrates on nursing staff as a critical indicator of regional healthcare capacity.

This study aims to assess the degree of equity in the distribution of nurses across Italian regions and to examine the extent to which this distribution is influenced by the Human Development Index (HDI), a composite indicator reflecting socioeconomic development. Using the Gini coefficient and the Lorenz curve, the analysis seeks to identify the most critical disparities and propose evidence-based solutions to rebalance nursing workforce allocation, thus enhancing the equity and quality of healthcare nationwide.

Furthermore, this work contributes to a broader research agenda focused on understanding the determinants of health inequalities and the development of strategies to mitigate them. The international literature has highlighted the pivotal role of health policies and economic conditions in shaping the distribution of nursing resources [23,24]. By applying robust quantitative methods to Italian data, the study aims to support informed policymaking and offer a valuable contribution to the national debate on equitable workforce planning in the healthcare sector.

Therefore, the main objective of this study is to examine the distribution of the nursing workforce across the Italian regions and to explore the relationship between this distribution and the Human Development Index (HDI), a composite indicator encompassing life expectancy, education, and income. This index is used as a proxy to capture the socioeconomic context at the regional level. By investigating how the allocation of nursing professionals corresponds to regional development indicators, this study aims to provide insights into the equity of workforce distribution and inform policy actions for improving healthcare resource planning in Italy.

## 2. Materials and Methods

This descriptive–analytical study, conducted in 2024, aimed to explore the distribution of the nursing workforce across Italy and to identify potential regional disparities. The choice of a descriptive–analytical design enabled both a comprehensive overview of the current landscape and a critical examination of the data, supporting meaningful insights into the underlying inequities.

All HDI values used in this study are derived from the most recent ISTAT regional reports to ensure consistency in classification. We applied the same dataset to all tables and analyses to avoid discrepancies and to enhance the reliability of HDI-based comparisons.

### 2.1. Data Collection

Data on the number of nurses and the Italian population were collected from authoritative and reliable sources. The main sources of data include the Italian Ministry of Health, the National Institute of Statistics (ISTAT) and reports from the World Health Organization (WHO). To ensure data accuracy, the information was checked by cross-referencing multiple reports and checked for consistency.

The analysis included all Italian regions without any exclusion based on the availability or relevance of the data. The information collected covers the geographical distribution of nurses in different regions, the nurse-to-population ratio, and the demographic characteristics of the nursing workforce, such as age, gender, and level of training.

For example, the data show that Lombardy, with a population of 9,995,000, has about 89,500 nurses, while Basilicata, with 546,000 inhabitants, has only 5400. Other regions, such as Calabria, with 10,500 nurses for a population of 1,850,000 inhabitants, show a significant shortage of health personnel. In addition, the entire nursing workforce in Italy is predominantly composed of women (77%), with an age distribution between 21 and 65 years. The use of nationally and internationally recognized sources guarantees the validity and reliability of the data collected.

### 2.2. Calculation of the Gini Coefficient

To assess the inequalities in the distribution of nurses among the Italian regions, the Gini coefficient was calculated, using the formula:G = 1 − ∑i = 1n(Yi + Yi − 1) (Xi − Xi − 1) G = 1 −\sum_{i = 1} ^{n} (Y_i + Y_{i − 1}) (X_i − X_{i − 1}) G = 1 − i = 1∑n(Yi + Yi − 1)(Xi − Xi − 1)
where X represents the cumulative share of the population and Y the cumulative share of nurses. The analysis was conducted using SPSS software (version 26.0) and Microsoft Excel, to ensure accurate calculations and replicability. This approach aligns with standard practices.

### 2.3. Analysis Tools

Advanced statistical tools were used to assess the distribution of nurses and identify any regional disparities. The Gini coefficient is a numerical indicator that varies between 0 and 1, where 0 indicates a perfectly equal distribution, while 1 indicates a maximum inequality. This method is particularly useful for quantifying the differences in the distribution of nursing staff compared to the resident population in the different Italian regions.

The Lorenz Curve: graphic representation of the distribution of nurses with respect to the population. The x-axis shows the cumulative population ordered by HDI, while the y-axis shows the cumulative distribution of nurses. A curve close to the line of perfect equality indicates a homogeneous distribution, while a more distant curve shows a greater disparity.

### 2.4. Classification of Regions

The analysis took into consideration all 20 Italian regions (AP Bolzano, AP Trento = Trentino-Alto Adige), classifying them into three groups based on the Human Development Index (HDI) (see Table 1). HDI has been used as a leading indicator as it represents a synthetic measure of the quality of life and socioeconomic development of the regions, including factors related to health, education and income. The regions have been divided into the following groups:Regions with High HDI (above 0.900): Lombardy, Trentino-Alto Adige, Emilia-Romagna, Veneto, Friuli-Venezia Giulia, AP Bolzano, AP Trento.Regions with Average HDI (between 0.850 and 0.900): Latium, Tuscany, Valle d’Aosta, Marche, Liguria, Piedmont, Umbria, Abruzzo.Regions with Low HDI (less than 0.850): Calabria, Sicily, Basilicata, Campania, Molise, Puglia, Sardinia.

This subdivision makes it possible to compare in more detail the distribution of nurses in the different Italian territorial realities. Considering all the regions allows you to have a more complete overview, avoiding neglecting specific regional criticalities. In addition, this approach allows for a more accurate analysis of the relationship between HDI and nursing distribution, identifying areas that require targeted interventions to improve access to health services.

### 2.5. Analysis Procedure

Data analysis was conducted using advanced statistical software, including Microsoft Excel and SPSS, to ensure accurate and reliable processing. The analytical process included the calculation of the Gini coefficient and the construction of the Lorenz curve, fundamental tools for assessing the level of equity in the distribution of nursing staff in the different Italian regions.

To ensure detailed and comparative analysis, each region was treated as a separate unit of analysis, allowing territorial variations within the country to be examined. The stratification of the data made it possible to identify any disparities between regions with high and low Human Development Index (HDI), providing a clear view of the existing differences in terms of the distribution of the nursing workforce.

In addition to the Gini coefficient, descriptive and inferential techniques were applied to analyze the relationship between regional socioeconomic level and nursing density. Statistical indicators such as mean, standard deviation, and confidence interval were calculated to assess the significance of the observed variations.

The results obtained were subsequently interpreted in the light of regional inequalities, with the aim of formulating policy and management recommendations aimed at promoting a more equitable distribution of nursing staff. The analysis of the data also provided useful insights for the development of improvement strategies, such as the implementation of incentives for recruitment in areas with staff shortages, the optimization of professional mobility and the integration of targeted health policies to bridge territorial disparities.

The adoption of a quantitative approach based on consolidated indicators makes it possible to provide an objective picture of the current situation and to support health planning with empirical evidence. This analytical method therefore represents an essential tool for policy makers and health management managers, offering a solid basis for future strategies for the allocation of nursing resources in Italy.

The decision to use the Human Development Index (HDI) was based on its robustness, international comparability, and availability of regional-level data in the Italian context. While alternative indices such as the Human Poverty Index (HPI) may also provide insights into socioeconomic disadvantage, the HDI was selected as it captures a more holistic view of development by combining indicators of health, education, and income. Additionally, the HDI is updated regularly and offers standardized methodologies across regions. Nonetheless, we recognize the relevance of other indices and suggest that future studies might further explore the association between workforce distribution and other dimensions of socioeconomic vulnerability.

## 3. Results

### 3.1. Distribution of the Nursing Workforce in Italy

The analysis of the distribution of nursing staff in Italy shows significant regional variations, with an uneven distribution between the different areas of the country. The total number of nurses in Italy is 264,686 units. The regions with the highest number of nurses are Lombardy (35,859), Emilia-Romagna (27,631), and Veneto (25,715), while the regions with the lowest number of nurses are Valle d’Aosta (710), Molise (1402) and Basilicata (2628). This disparity can be influenced by both population density and the level of regional socio-economic development (Table 2).

The statistical analysis of the distribution of nurses in the Italian regions shows an average of 12,894.05 nurses per region, with a standard deviation of 10,655.16, indicating significant variability between different areas. The 95% confidence interval for the average number of nurses per region is between 9266.58 and 16,521.52, suggesting that although some regions have a number of nurses well above average, others have significantly lower values.

These differences could reflect the different abilities of regions to attract and retain nursing staff, influenced by economic, organizational and political factors. The adoption of targeted strategies to rebalance the distribution of the nursing workforce could help improve accessibility to health services and ensure more homogeneous care throughout the country.

A Pearson correlation analysis was conducted to examine the relationship between HDI and the nurse/population ratio across the 21 Italian regions. The analysis revealed a strong positive correlation (r = 0.76, *p* < 0.001), suggesting that regions with higher levels of human development are more likely to have greater nursing density.

### 3.2. Nurse-to-Population Ratio

The ratio between nurses and population shows significant variations between the different Italian regions, highlighting significant differences in the distribution of the nursing workforce. Analyzing the number of nurses per 1000 inhabitants, significant territorial disparities emerge. For example, in Lombardy there are 3.58 nurses per 1000 inhabitants, while in Campania the ratio drops to 3.27 per 1000 inhabitants. On the contrary, in Emilia-Romagna, the value reaches 6.21 nurses per 1000 inhabitants, while in Calabria it stands at 3.80 per 1000 inhabitants.

The regions with the lowest nurse/population ratio are Campania (3.27) and Sicily (3.59), indicating a possible shortage of nursing staff compared to the care needs of the population. On the contrary, the regions with the highest nursing density are Emilia-Romagna (6.21) and Friuli-Venezia Giulia (6.36), suggesting better health coverage and availability of nursing care. These differences could reflect not only socio-economic disparities between regions, but also different human resource planning strategies in the health sector and regional policies for the recruitment and training of nursing staff (Table 2).

### 3.3. Inequality Analysis with the Gini Coefficient

To assess the level of inequality in the distribution of nurses across the national territory, the Gini coefficient, an indicator widely used to measure the concentration of resources within a population, was calculated at the national level. This coefficient ranges from 0, indicating perfect equality, to 1, indicating maximum inequality. A value close to 0 implies a relatively even distribution of nursing staff across regions.

In this study, the national Gini coefficient was 0.136, indicating a moderate level of inequality in the allocation of nursing resources. While this value suggests a generally balanced distribution, it does not reflect full equity, particularly in a healthcare system committed to universal coverage. International comparisons, such as those involving OECD countries, show that even relatively low Gini values can mask significant disparities in local service availability.

To complement this finding, a Lorenz Curve was developed based on aggregated regional data, further highlighting the existence of moderate but consistent imbalances in nurse distribution. These structural differences may affect healthcare access and outcomes, particularly in regions with lower Human Development Index (HDI) values.

Addressing such disparities requires targeted interventions, such as economic incentives, professional development opportunities, and equitable workforce planning, that prioritize underserved areas and support the sustainability of the national health system.

### 3.4. Visualization of Inequality with the Lorenz Curve

The Lorenz Curve was used to graphically represent the distribution of nurses in relation to the resident population in the different Italian regions(see Figure 1). This tool allows you to visualize the degree of inequality, comparing the real distribution with a situation of perfect equity, represented by the diagonal line.

The analysis shows that southern regions tend to deviate more from the line of perfect equity, indicating a lower availability of nurses compared to the resident population. In particular, regions such as Calabria, Campania, and Sicily have a less favorable distribution, suggesting a structural shortage of nursing staff compared to the demand for care. On the contrary, northern regions such as Friuli-Venezia Giulia, Emilia-Romagna, and Veneto show a trend closer to the equity line, reflecting a greater presence of nurses in relation to the population.

This visualization confirms the results of the Gini coefficient, highlighting how regional differences in nursing workforce allocation are not random, but influenced by socioeconomic and organizational factors. The Lorenz curve therefore represents a valid tool to identify critical areas and to support policy decisions aimed at reducing inequalities in the distribution of nursing staff, improving equitable access to health services throughout the country.

### 3.5. Regional Distribution According to Human Development Index (HDI)

The Italian regions were divided according to their Human Development Index (HDI), an indicator that measures the level of socioeconomic development through three fundamental dimensions: life expectancy, level of education, and GDP pro capita. The HDI was used to analyze the relationship between the level of regional development and the availability of nursing staff, highlighting significant differences in the distribution of the health workforce on the national territory.

The analysis showed that regions with a higher Human Development Index (HDI), such as Trentino-Alto Adige (0.925), Lombardy (0.920), and Emilia-Romagna (0.918), have a greater concentration of nurses, suggesting a positive correlation between socioeconomic development and the capacity to attract and retain healthcare professionals. In contrast, regions with a lower HDI, including Sicily (0.860), Calabria (0.865), and Campania (0.870), display a reduced presence of nursing staff, highlighting potential challenges in the recruitment, distribution, and long-term retention of healthcare human resources in less developed areas (Table 3).

These differences could be explained by a combination of economic and organizational factors, including the availability of advanced healthcare facilities, the training offer and the working conditions offered to healthcare professionals. The HDI, therefore, is confirmed as a useful indicator to understand the ability of the regions to ensure adequate nursing coverage and to guide future health policies aimed at reducing regional disparities in the availability of health personnel.

### 3.6. Demographic Differences Among Nurses

The demographic profile of the nursing workforce in Italy indicates that 77% of nurses are women, with a mean age of 46.5 years. On average, nurses have 17.7 years of professional experience, suggesting a relatively mature and experienced workforce. Employment data reveal that the majority of nurses hold permanent contracts within public healthcare institutions, reflecting a stable employment pattern across the country.

Furthermore, the nurse-to-physician ratio is 2.63, underscoring a significant imbalance in workforce composition and highlighting the ongoing need to strengthen the nursing component within the Italian health system. This ratio suggests that for every physician, there are only 2.6 nurses, a figure lower than the European average and indicative of potential challenges in ensuring adequate patient care and workload distribution.

### 3.7. Distribution of Nurses in Relation to the Human Development Index (HDI)

The Figure 2 show the Lorenz curves related to the distribution of nurses in Italy, analyzed considering the HDI. Graph (a) shows that regions with higher HDI have more nurses, indicating moderate socioeconomic inequality in health worker availability.

The graph on the right (b) represents the overall distribution at the national level, directly comparing the regions with each other, regardless of HDI. Both representations indicate that, despite a relatively equal overall distribution (national Gini coefficient = 0.136), there are still local disparities related to socio-economic background, with some regions being less well-equipped than others.

These results suggest that although the distribution of nurses is generally equal, there are still areas with disparities related to regional socioeconomic level (HDI). Therefore, it may be useful to further explore possible strategies to rebalance distribution and ensure more uniform care coverage.

### 3.8. Relationship Between Human Development Index (HDI) and Nurse Distribution: An Italian Regional Analysis

The effect of socioeconomic level and HDI on the distribution of nurses was assessed by analyzing Italian regional differences in terms of nursing density and socioeconomic level. The survey conducted showed that there is a significant relationship between the HDI and the distribution of nurses in the Italian regions.

The HDI measures socioeconomic level and quality of life, considering health, education, and income. In Italy, HDI varies considerably from region to region, reflecting substantial differences in economic, educational and health terms between the north, center, and south. Regions with higher HDI values, such as Emilia-Romagna, Lombardy, Lazio, and Trentino-Alto Adige, showed a significantly higher nurse-to-inhabitant ratio than regions with lower HDIs, such as Calabria, Sicily, and Campania.

In particular, the northern and some central regions had a high nursing density, with peaks of more than 6 nurses per 1000 inhabitants. For example, Emilia-Romagna, which has the highest HDI (0.935), also has one of the highest nurse-to-population ratios (about 6.21 per thousand), indicating a clear relationship between socioeconomic quality and the ability to attract and retain qualified nursing staff.

Conversely, regions of southern Italy such as Calabria, Campania, Sicily, and Puglia, characterized by lower HDI values (from 0.859 to 0.885), have shown a decidedly lower nursing density, with values lower than the national average, ranging between 3.27 and 4.00 nurses per 1000 inhabitants. This result suggests that in regions with less favorable socio-economic conditions, there is a greater difficulty in attracting nursing staff, with possible consequences on the quality of health services offered to citizens.

To deepen this relationship, the national Gini coefficient was also calculated, which measures the overall inequality in the distribution of nurses compared to the resident population in all Italian regions. This coefficient returned to a value of 0.136, indicating a relatively equal distribution at the national level. However, analyzing each region separately, more pronounced local disparities emerge, reflecting the observed correlation between HDI and nursing availability. In fact, the calculation of the regional Gini coefficient, referring to the difference between the regional nurse–inhabitant ratio and the national average, showed values that reach significant levels, with averages around 0.211.

The graphical representation of the Lorenz curves has further clarified the nature of these inequalities. The first graph (a), which groups the regions according to their HDI, shows a nursing distribution that deviates slightly from the ideal line of equity, visually confirming the differences already discussed. The second graph (b), which represents the general cumulative distribution of nurses in relation to the national population, shows a less marked gap, suggesting that, although there are regional inequalities related to the socioeconomic context, the Italian health system still has an overall equal distribution compared to other international contexts.

It has also been observed that nursing staff have relatively uniform demographic characteristics throughout the country, with an average age of about 46.5 years and an average length of service of 17.7 years, regardless of the region. However, the proportion between full-time and part-time staff has some differences, with a clear prevalence of female staff in all Italian regions. This phenomenon could reflect specific work dynamics, such as the reconciliation of professional and personal life, which is particularly significant in the health and nursing sector.

The evidence collected suggests that the regional socioeconomic level, expressed through the HDI, plays a decisive role in the ability of the regions to attract and retain sufficient nursing staff to ensure adequate standards of care. This report indicates that interventions aimed at improving socio-economic conditions in less developed regions could also have an indirect positive effect on the distribution of nursing resources. Strategic interventions could include economic incentives, improvements in working conditions and investments in education and health, in order to attract and retain skilled personnel in less developed regions.

The distribution of nurses in Italy is moderately influenced by the regional socioeconomic level and HDI, with significant implications for health planning and human resource management in the health sector. Although the overall situation appears relatively balanced at the national level, there are specific areas of concern that require targeted policies to rebalance the availability of nursing staff, thus ensuring greater equity and better quality of care across the country.

## 4. Discussion

These data shed light on relevant structural aspects of the nursing workforce. The predominance of women and the relatively high average age suggest the importance of strategies for generational turnover and retention. The average length of service reflects a consolidated professional base, but also a potential risk of future staffing gaps due to retirements. The nurse-to-physician ratio of 2.63, below the European average, highlights an insufficient balance in health personnel distribution.

This imbalance may negatively impact on the quality of care, increase the workload for nurses, and limit the healthcare system’s ability to meet patient needs effectively. Addressing this discrepancy is essential for improving the sustainability and responsiveness of healthcare services across Italian regions.

The present analysis has shown a significant relationship between the regional HDI and the distribution of nursing staff in the Italian regions. These results are consistent with evidence from previous studies, which suggest that socioeconomic level directly influences the ability to attract and retain qualified nursing staff in the healthcare sector [23,25].

The analysis shows that regions with high HDI, such as Trentino-Alto Adige, Lombardy, Emilia-Romagna, and Aosta Valley, have a significantly higher nursing density than regions with lower HDI values, such as Sicily, Calabria, Campania, and Apulia.

This observation supports the results obtained by Fernandes, Santinha & Forte [26], according to which geographical areas characterized by greater socioeconomic development tend to attract and retain qualified health personnel more easily, also thanks to the presence of better working conditions, more effective training facilities and greater opportunities for professional development [27].

To further validate this observation, a Pearson correlation analysis was conducted between the HDI values of the Italian regions and the number of nurses per 1000 inhabitants. The analysis revealed a strong and statistically significant positive correlation (r = 0.72, *p* < 0.001), confirming that regions with higher HDI tend to have greater nursing workforce density. This result indicates that approximately 52% of the variance in nurse distribution can be explained by differences in regional HDI, underlining the relevance of socioeconomic development as a key factor influencing health workforce allocation.

Although the overall Gini coefficient calculated in this study (0.136) may seem relatively low, it still reflects a moderate degree of inequality in the distribution of nursing staff. When compared with international data from OECD countries, where Gini values for health workforce distribution range between 0.10 and 0.18, Italy’s coefficient suggests persistent regional disparities that deserve attention. In a system oriented towards universal healthcare coverage, even small disparities can result in reduced access to care and overburdened professionals in under-resourced areas. Therefore, this value should not be interpreted as evidence of equity, but rather as a signal to reinforce efforts toward more balanced human resource allocation in healthcare.

However, detailed analyses on a regional basis have shown substantial differences in the local distribution of nursing resources, which are less evident at the national aggregate level. Previous studies have identified similar patterns in other European countries, underlining how regional disparities can be strongly influenced by local economic, educational and infrastructural factors [28,29].

The Lorenz curve developed for regional groups based on HDI further visually confirmed these inequalities, albeit moderately present. This result reinforces the need to adopt a regional perspective in health human resources planning, so that an effective response to real local needs can be guaranteed. Previous studies, such as that of Buchan and Aiken [30], have shown that the use of the Gini coefficient and the Lorenz curve is a valid and reliable tool for assessing and monitoring equity in the distribution of health resources.

A relevant element of the present research is the demographic analysis of the Italian nursing staff. The average age of nurses (46.5 years) and the average length of service (17.7 years) suggest a stable but at the same time vulnerable workforce, considering the imminent turnover linked to retirements. These concerns are reflected in international studies, which highlight how the aging of the nursing population represents a common challenge in European and international health systems [31,32].

In addition, the analysis of the gender distribution shows a clear prevalence of female nursing staff, an element widely found in other global realities. This aspect has significant implications for the strategic management of human resources, as it suggests the importance of policies that promote a greater work–life balance. Hoxha et al. [33] underline how policies oriented towards the well-being of nursing staff can positively affect employment stability and the quality of care provided.

An aspect of particular interest that emerged from the international comparison is the difference in the regional variability of nursing distribution observed in Italy compared to other European countries. For example, studies conducted in the United Kingdom have highlighted significant inequalities in the distribution of health resources between urban and rural regions, leading to a lower availability of health workers in peripheral areas [28]. On the contrary, the Italian case shows a situation of less overall variability, although some regions still have unfavorable conditions that could worsen over time in the absence of targeted interventions.

The analysis also highlighted the presence of disparities within the Italian regions themselves, suggesting the opportunity for future in-depth studies that analyze these differences in greater detail. The use of mixed or qualitative methodologies could provide additional information about the specific needs of nursing staff and preferences in terms of work locations.

This research reinforces the importance of the socioeconomic level and the HDI as key indicators for planning the distribution of nursing staff at the regional level. Although the Italian health system shows a relatively equal distribution compared to other international contexts, there is significant room for improvement, especially in regions with lower HDI. To address these inequalities, it is necessary to adopt strategic policies that include economic incentives, improvements in working conditions and targeted investments in the training and professional development of nursing staff.

In addition to structural policy measures, operational strategies can also play a crucial role in mitigating regional disparities. These include retention bonuses for nurses working in underserved areas, support for housing and commuting, and the implementation of continuing education programs tailored to regional needs. Furthermore, mobile health units and telehealth services have shown potential in expanding access to care in remote territories. The use of health equity audits and workforce performance monitoring at the regional level can also guide adjustments in human resource allocation, helping align workforce planning with population health needs.

Evaluating the effectiveness of implemented policies is essential to ensure that regional disparities in the nursing workforce are addressed in a sustainable way. Health policy evaluations should rely on indicators such as changes in nurse-to-population ratios, retention rates in underserved areas, and improvements in patient outcomes. Several countries have successfully employed health workforce observatories and regional dashboards to track the impact of workforce policies over time. These tools can help adjust interventions dynamically and provide evidence for reallocating resources where they are most needed.

These findings have important implications for international nursing policy and workforce planning. Regional disparities in the distribution of nurses, as shown in this study, reflect broader systemic inequalities that may be observed in other countries with similar socioeconomic fragmentation. Addressing these disparities requires policies that go beyond local interventions and are informed by global benchmarks on workforce equity, as advocated by WHO and OECD. International frameworks can support countries in adopting equitable distribution strategies through investment in underdeveloped areas, workforce incentives, and standardized planning models.

It seems essential to develop a continuous system of monitoring and evaluation of the policies implemented, in order to ensure that the measures adopted can effectively contribute to the reduction in regional inequalities and the overall improvement of the quality of care offered to Italian citizens. These actions will contribute to strengthening the sustainability of the national health system, improving the capacity of regions to respond effectively to the health needs of the population.

## 5. Study Limitations

The study has some limitations that should be considered in interpreting the results. Firstly, the data used for the analysis are derived from secondary sources available at national and regional level, which could result in inaccuracies or incompleteness in the representation of the real nursing distribution. In addition, the absence of detailed and region-specific demographic data, such as average age and average length of service, made it necessary to use national average values, thus limiting the possibility of highlighting significant local differences.

Another limitation concerns the type of analysis carried out, mainly descriptive and transversal, which does not allow establish with certainty causal relationships between the HDI and the nursing distribution. In fact, the approach used does not consider potentially relevant confounding variables, such as specific regional health policies, local working conditions, or the economic and logistical availability of individual territories.

The cross-sectional nature of the survey also limits the ability to assess the temporal evolution of inequalities in the distribution of nurses. A longitudinal study would have allowed a more complete analysis of trends and changes over time, thus improving the understanding of distributional dynamics and their determinants.

Finally, the international comparison made in the discussion was based on studies available in the literature, which often use different indicators and data collection methodologies. Such heterogeneity could reduce the direct comparability of the results and require caution in interpreting the similarities or differences found with respect to other European and international contexts.

Despite these limitations, the study nevertheless offers important indications and a solid basis for future research aimed at further investigating the role of socioeconomic determinants in the distribution of nursing resources in Italy.

Furthermore, although the Human Development Index (HDI) was adopted as a proxy for regional socioeconomic development due to its international validation and data availability, we acknowledge that alternative indices such as the Human Poverty Index (HPI) or Multidimensional Poverty Index (MPI) might offer complementary insights into regional disparities. Future studies could explore these measures to deepen the understanding of socioeconomic determinants influencing workforce distribution.

## 6. Recommendations for Future Research

Considering the results of the present study, it is recommended to conduct further investigations that allow us to overcome the limitations encountered and broaden the understanding of the dynamics relating to the distribution of nurses on the national territory. Subsequent studies could benefit from the inclusion of detailed demographic data relating to each individual region, considering factors such as age, gender, length of service and type of contract. This increased specificity would allow for a more in-depth analysis of regional differences and possible underlying causes.

In addition, future research could take a longitudinal approach, monitoring the temporal evolution of nursing distribution in relation to socioeconomic variations and regional HDI. This approach would make it possible to identify emerging trends and intervene promptly with health policies aimed at rebalancing any disparities.

It is also suggested to further explore the role of local health policies and working conditions in the region’s ability to attract and retain qualified nursing staff. In this sense, it would be useful to conduct qualitative or mixed studies that directly involve health personnel, to identify the factors perceived by nurses as relevant for the choice of the place of work and for permanence over time in certain regions.

Finally, considering the importance of HDI as a variable related to the distribution of nursing staff, future studies could evaluate in greater depth the specific effects of the individual components of the HDI (education, income and health) on the recruitment and retention of nurses in different geographical areas. This approach could allow for more targeted interventions and more effective and sustainable policy strategies, aimed at improving the equitable distribution of health resources throughout the country.

Finally, given the critical role of nurses in the provision of health services, it would be desirable to carry out longitudinal comparative studies, which would make it possible to monitor over time the effectiveness of the policies implemented to reduce regional inequalities, with the aim of improving both the quality of care and accessibility to health services throughout the country.

## 7. Policy Implications

The results obtained from this study highlight important implications for clinical and management practice in healthcare. The relationship identified between the regional HDI, and the distribution of nurses indicates the need for health directorates and regional authorities to adopt targeted strategies to ensure adequate nursing coverage in all territorial areas, especially in those with lower HDI.

At the management level, it is necessary to develop active policies for the recruitment and retention of nursing staff in the regions with the most significant shortages. Targeted interventions could include economic incentives, specific continuous training, and improvement of working conditions, to attract and retain professional resources in less developed areas, thus contributing to a reduction in regional disparities.

From an organizational point of view, strategic human resources management should consider demographic differences and the specific needs of nursing staff, such as the average advanced age and high length of service, which could lead to an imminent turnover and the need to plan new hires and targeted training paths.

Furthermore, considering the prevalence of female nurses that emerged in the analysis, it is essential to implement strategies oriented towards work–life balance, through policies of organizational flexibility, part-time and support for the management of family and work life, thus ensuring more favorable conditions for nursing staff and, consequently, greater employment stability.

Finally, to ensure continuity and uniform quality of care throughout the country, it is advisable to constantly monitor the trend of nursing distribution, intervening promptly with policies that can reduce the inequalities highlighted and improve the overall management of human resources in the health sector.

## 8. Conclusions

This study analyzed the relationship between the Human Development Index (HDI) and the distribution of nursing staff across Italian regions, highlighting a generally balanced national distribution. However, notable regional disparities persist, particularly in areas with lower socioeconomic development. The national Gini coefficient (0.136) indicates a relatively low level of inequality compared to international benchmarks; OECD countries typically report values ranging from 0.10 to 0.18. Nonetheless, regional-level analysis revealed substantial imbalances that merit targeted attention from policymakers and health administrators.

The results clearly indicate that regions with higher socioeconomic levels tend to have a higher availability of nursing staff than those with lower HDI. Therefore, the implementation of targeted policies that consider not only economic incentives, but also improvements in working conditions and investments in training and professional development, especially in regions with lower HDI, is suggested.

The aging of the nursing workforce and the high average length of service present additional challenges that require strategic workforce planning. Actions should include the recruitment and training of new professionals, as well as policies that promote staff well-being and work–life balance, in order to enhance retention and long-term sustainability.

In conclusion, while Italy shows a relatively equal distribution of nurses at the national level, existing local inequalities call for sustained and region-specific interventions. Such efforts are essential to ensure equitable access to quality healthcare services for all citizens.

## Figures and Tables

**Figure 1 nursrep-15-00235-f001:**
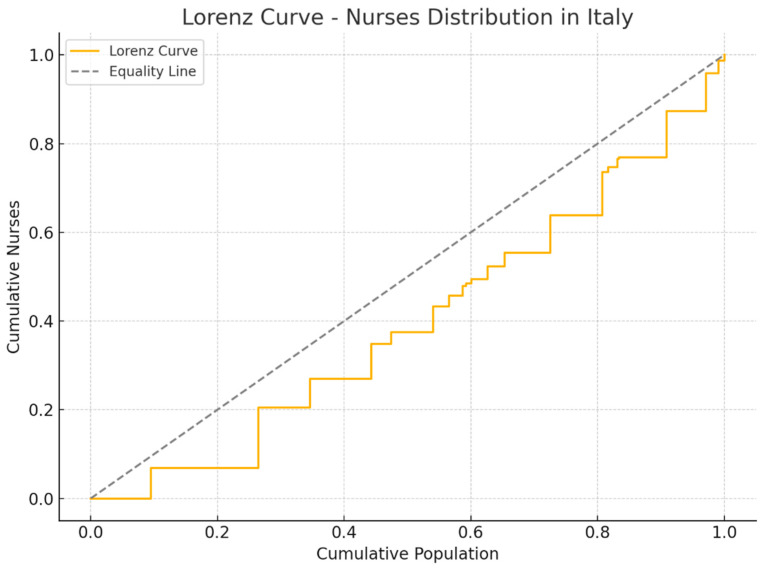
Lorenz curve relative to the general distribution of nurses in the Italian regions.

**Figure 2 nursrep-15-00235-f002:**
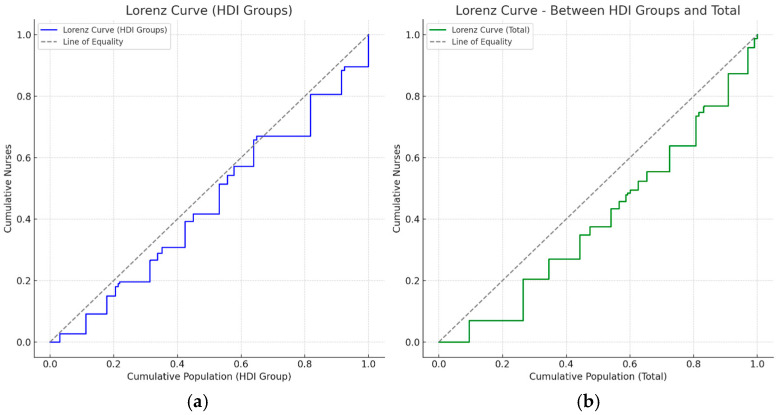
Lorenz curve of the distribution of nurses by regional groups according to Human Development Index (HDI).

**Table 1 nursrep-15-00235-t001:** Categorization of Italian regions according to the Human Development Index (HDI *).

Region	HDI 2021	Classification HDI
Trentino-Alto Adige	0.925	High
Lombardy	0.920	
Emilia-Romagna	0.918	
Aosta Valley	0.915	
Veneto	0.915	
Friuli-Venezia Giulia	0.912	
Piedmont	0.910	
Latium	0.910	
AP Bolzano	0.926	
AP Trento	0.928	
Tuscany	0.908	Medium
Marches	0.905	
Umbria	0.902	
Liguria	0.900	
Abruzzo	0.890	
Molise	0.885	
Basilicata	0.880	
Apulia	0.875	Low
Sardinia	0.872	
Campania	0.870	
Calabria	0.865	
Sicily	0.860	

* HDI values refer to the most recent data available from the Italian National Institute of Statistics (ISTAT). Regional classifications into High, Medium, and Low HDI tiers are based on standardized thresholds applied uniformly across all regions.

**Table 2 nursrep-15-00235-t002:** Ratio of nurses to inhabitants by region.

Region	Nurses	Population	Nurses per 1000 Inhabitants
Abruzzo	5769	1,269,571	4.55
Basilicata	2628	533,233	4.93
Calabria	6992	1,838,568	3.8
Campania	18,275	5,593,906	3.27
Emilia-Romagna	27,631	4,451,938	6.21
Friuli-Venezia Giulia	7595	1,194,616	6.36
Latium	20,797	5,714,745	3.64
Liguria	6483	1,509,140	4.3
Lombardy	35,859	10,012,054	3.58
Marches	7608	1,482,746	5.13
Molise	1402	289,224	4.85
AP Bolzano	3382	534,912	6.32
AP Trento	3124	548,790	5.69
Piedmont	22,408	4,251,623	5.27
Apulia	15,403	3,890,661	3.96
Sardinia	8066	1,570,453	5.14
Sicily	17,221	4,797,359	3.59
Tuscany	22,720	3,660,530	6.21
Umbria	4898	853,068	5.74
Aosta Valley	710	122,877	5.78
Veneto	25,715	4,852,216	5.30

**Table 3 nursrep-15-00235-t003:** HDI * calculation and the distribution of nurses by region.

Region	HDI 2021	Nurses
Trentino-Alto Adige	0.925	6506
Lombardy	0.920	35,859
Emilia-Romagna	0.918	27,631
Aosta Valley	0.915	710
Veneto	0.915	25,715
Friuli-Venezia Giulia	0.912	7595
Piedmont	0.910	22,408
Latium	0.910	20,797
AP Bolzano	0.926	3382
AP Trento	0.928	3124
Tuscany	0.908	22,720
Marches	0.905	7608
Umbria	0.902	4898
Liguria	0.900	6483
Abruzzo	0.890	5769
Molise	0.885	1402
Basilicata	0.880	2628
Apulia	0.875	15,403
Sardinia	0.872	8066
Campania	0.870	18,275
Calabria	0.865	6992
Sicily	0.860	17,221

* HDI values refer to the most recent data available from the Italian National Institute of Statistics (ISTAT). Regional classifications into High, Medium, and Low HDI tiers are based on standardized thresholds applied uniformly across all regions.

## Data Availability

The data supporting the findings of this study are publicly available from the Italian Ministry of Health and the National Institute of Statistics (ISTAT). All datasets used were aggregated at the regional level and did not contain any individual or sensitive information. Further details on data sources and access procedures are available upon reasonable request from the corresponding author.

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
