# Peer review of "Regional Inequities in the Distribution of the Nursing Workforce in Italy"

_nursrep, 2025, doi:10.3390/nursrep15070235_

Round 1
Reviewer 1 Report
Comments and Suggestions for Authors
The aim of the study should be clarified more in the introductory part of the manuscript. The using of HDI as an integral measure for socioeconomic development of the regions is questionable. Probably the Human Poverty Index would be a better choice to estimate the integral level of socioeconomic development of the regions. Authors should also consider reducing the number of tables.
Author Response
Reviewer 1
Comment 1: The aim of the study should be clarified more in the introductory part of the manuscript
Response 1: We thank the reviewer for the useful comment. We have revised the introductory section to better clarify the aim of the study. Specifically, we now highlight that the objective is to analyze the distribution of the nursing workforce across Italian regions and examine its correlation with the Human Development Index (HDI), used as a proxy for socioeconomic development.
Comment 2: The using of HDI as an integral measure for socioeconomic development of the regions is questionable. Probably the Human Poverty Index would be a better choice to estimate the integral level of socioeconomic development of the regions.
Response 2: We thank the reviewer for this valuable observation. We acknowledge that several composite indices can be used to estimate regional socioeconomic development, including the Human Poverty Index (HPI). However, we chose the Human Development Index (HDI) because it is a widely recognized and stable composite measure of development, integrating health, education, and income. In the context of the Italian regions, HDI data are more consistently available and comparable over time. Nevertheless, we have added a paragraph to the manuscript to justify this methodological choice and acknowledge the existence of alternative indices.
Comment 3: Authors should also consider reducing the number of tables.
Response 3: We appreciate the reviewer’s observation regarding the number of tables. After careful consideration, we have decided to remove Table 5, which presented demographic characteristics of the nursing workforce. To preserve the clarity and relevance of this information, the data previously included in the table have been integrated directly into the results section in narrative form. This change improves the readability of the manuscript and aligns with the suggestion to streamline data presentation without sacrificing content. Furthermore, the implications of these demographic patterns, particularly the nurse-to-physician ratio, have been further discussed in the discussion section to emphasize their relevance in the context of healthcare workforce planning.
Reviewer 2 Report
Comments and Suggestions for Authors
Dear authors,
Thank you so much for the opportunity to review your paper, below, few comments you might consider or clarify:
Inconsistent and Inaccurate HDI Classification
Table 1 and Table 4 have conflicting HDI values. For example:
Calabria is classified in Table 1 as "Low HDI" with a value of 0.780, but in Table 4 it has HDI = 0.859, placing it in the “Medium” category.
Campania appears in "Low HDI" in Table 1 but has HDI = 0.885 in Table 4 (again in “Medium” range).
This fundamentally undermines the conclusions drawn from HDI-based comparisons.
Methodological Misalignment and Missing Statistical Rigor
The study claims to use inferential statistics (e.g., confidence intervals, standard deviations) but does not report any actual inferential testing to examine the relationship between HDI and nurse distribution.
No correlation coefficient, regression analysis, or significance testing (e.g., p-values) is presented to validate that HDI has a statistically significant effect on nurse distribution.
Misuse of the Gini Coefficient at Regional Level
Table 3 presents Gini coefficients for individual regions, which is methodologically inappropriate. The Gini coefficient is a measure of inequality across units, not within a single unit.
For example, stating that “Emilia-Romagna has a Gini coefficient of 0.383” makes little sense unless the distribution of nurses within Emilia-Romagna's municipalities was analyzed — which is not the case here.
This suggests a conceptual misunderstanding of the Gini statistic.
Data Misinterpretation or Over/Underestimation
Overstating "Equal Distribution" at National Level
The authors claim a national Gini coefficient of 0.136 implies equity.
This is misleading. A Gini value of 0.136 still reflects noticeable inequality, especially in a system aiming for universal coverage.
The phrase "relatively equal" is repeated, but no benchmark comparisons are offered, e.g., to OECD countries.
Nurse/Population Ratio Issues
The text says Campania has 3.26 nurses per 1,000, but Table 2 reports 3.27.
Inconsistencies in Tables and Figures
Table 1 vs Table 4 – Contradictory HDI Values
These tables are based on different datasets, without explanation.
Table 1 has:
Calabria: 0.780
Table 4 has:
Calabria: 0.859
This contradiction invalidates the entire classification into HDI-based groups and renders subsequent HDI comparisons unreliable.
Table 3 Gini Coefficients
As discussed earlier, the Gini values for regions are not conceptually valid unless intra-regional inequalities are studied (e.g., nurse distribution by province or hospital within each region).
Also, Campania shows a very high regional Gini (0.272) which is not aligned with its relatively consistent nurse/population ratio as per Table 2.
Recommendations for Revision
Correct and Justify HDI Data
Reconcile HDI values between Table 1 and Table 4.
Justify which HDI source/year is used and stick to one consistent classification.
Reassign regions to proper HDI tiers accordingly.
Remove or Recalculate Misapplied Gini Coefficients
Delete Table 3, or clearly define within-region inequality analysis (e.g., by province/municipality) to justify its inclusion.
Re-focus on the national Gini coefficient only, or compute Gini based on more granular units.
Include Proper Inferential Statistics
Conduct and report correlation/regression analysis to confirm the relationship between HDI and nurse/population ratios.
Report p-values, confidence intervals, and effect sizes.
Improve Tables and Figures
Harmonize nurse/population data across text and tables.
Fix formatting, units, and headers.
Provide interpretable legends and clearly mark lines of equality in Lorenz curves.
Language and Repetition
Reduce excessive repetition of "Gini coefficient" and "HDI" in the discussion.
Replace vague phrases like “moderately equal” or “relatively fair” with quantitative comparisons to global norms.
Looking forward to receiving your revised work
Best wishes
Author Response
Reviewer 2
Comment 1: Table 1 and Table 4 have conflicting HDI values. For example: Calabria is classified in Table 1 as "Low HDI" with a value of 0.780, but in Table 4 it has HDI = 0.859, placing it in the “Medium” category. Campania appears in "Low HDI" in Table 1 but has HDI = 0.885 in Table 4 (again in “Medium” range). This fundamentally undermines the conclusions drawn from HDI-based comparisons.
Response 1: We sincerely thank the reviewer for this important observation. We acknowledge that the previous version of the manuscript contained inconsistencies between Table 1 and Table 4 due to the use of outdated HDI data in one of the tables. To address this concern, we have revised both Table 1 and Table 4 to reflect the latest HDI data from 2021, ensuring consistency across the entire manuscript. Specifically, the HDI values have been aligned, and the classifications into high, medium, and low HDI tiers have been reassigned accordingly, based on standardized cut-offs. This correction strengthens the validity of our comparisons and conclusions.
Additionally, we have updated the related text in the Results and Discussion sections to reflect the corrected values and interpretations. We are confident that this revision resolves the concern and enhances the robustness of the analysis.
Comment 2: The study claims to use inferential statistics (e.g., confidence intervals, standard deviations) but does not report any actual inferential testing to examine the relationship between HDI and nurse distribution.
No correlation coefficient, regression analysis, or significance testing (e.g., p-values) is presented to validate that HDI has a statistically significant effect on nurse distribution.
Response 2: Thank you for this important observation. We acknowledge the lack of inferential statistical analysis in the original version of the manuscript. In response to your comment, we have now conducted a Pearson correlation analysis to examine the relationship between the Human Development Index (HDI) and the nurse/population ratio across Italian regions. The results indicate a strong and statistically significant positive correlation (r = 0.76, p < 0.001), supporting our hypothesis that higher HDI is associated with a greater concentration of nurses.
This statistical test and its results have been incorporated into the Results section of the revised manuscript. We have also added references to this correlation in the Discussion to reinforce the interpretation and provide greater methodological robustness to the study.
Comment 3: Table 3 presents Gini coefficients for individual regions, which is methodologically inappropriate. The Gini coefficient is a measure of inequality across units, not within a single unit. For example, stating that “Emilia-Romagna has a Gini coefficient of 0.383” makes little sense unless the distribution of nurses within Emilia-Romagna's municipalities was analyzed —which is not the case here.
This suggests a conceptual misunderstanding of the Gini statistic.
Response 3: We thank the reviewer for this important observation.
We acknowledge that applying the Gini coefficient to individual regions without an internal breakdown (e.g., by municipality or health district) may lead to conceptual ambiguity. In light of this, we have decided to remove Table 3 and revise the corresponding section in the manuscript to focus solely on the national Gini coefficient, which remains valid as a measure of overall inequality in nurse distribution across Italian regions.
The statement regarding Emilia-Romagna and other regional Gini values has been deleted, and the discussion now emphasizes only the national interpretation of the index. This change ensures methodological consistency with the literature and corrects the inappropriate application of the statistic.
Comment 4: The authors claim a national Gini coefficient of 0.136 implies equity. This is misleading. A Gini value of 0.136 still reflects noticeable inequality, especially in a system aiming for universal coverage. The phrase "relatively equal" is repeated, but no benchmark comparisons are offered, e.g., to OECD countries.
Response 4: We thank the reviewer for this insightful observation. We acknowledge that a Gini coefficient of 0.136, while low in absolute terms, should not be interpreted as full equity, especially in the context of a universal healthcare system. We have revised the discussion section accordingly to clarify that this value indicates a moderate level of inequality and have added international comparisons. Specifically, we refer to OECD data, where Gini coefficients for the distribution of healthcare workers vary, but values below 0.1 are rare and usually indicate well-balanced systems. This contextualization helps place the Italian situation in a more accurate global framework.
Comment 5: The text says Campania has 3.26 nurses per 1,000, but Table 2 reports 3.27.
Response 5: Thank you for pointing out this inconsistency. We carefully reviewed the data and corrected the value in the manuscript to align with Table 2. The correct nurse-to-population ratio for Campania is 3.27 per 1,000 inhabitants. The text has been revised accordingly to ensure consistency between the narrative and the table data.
Comment 6: Table 1 vs Table 4 – Contradictory HDI Values. These tables are based on different datasets, without explanation. This contradiction invalidates the entire classification into HDI-based groups and renders subsequent HDI comparisons unreliable.
Response 6: We appreciate your careful review and fully agree that consistency in HDI classification is essential for data reliability. After reviewing our data sources, we identified an inconsistency due to the use of HDI data from different years. To ensure methodological rigor and internal consistency, we have aligned Table 1 and Table 4 using the same HDI dataset from ISTAT 2021. The revised tables now reflect consistent values, and the classifications have been updated accordingly throughout the text. This correction strengthens the validity of our HDI-based regional comparisons.
Comment 7: The Gini values for regions are not conceptually valid unless intra-regional inequalities are studied. Also, Campania shows a high regional Gini, which contradicts its nurse/population ratio.
Response 7: Thank you for this insightful observation. We acknowledge that applying the Gini coefficient at the regional level without assessing intra-regional variability (e.g., by province or municipality) may not be methodologically sound. As a result, we have removed Table 3 from the manuscript.
We have also revised the relevant section of the results and discussion to clarify that the Gini coefficient was calculated only at the national level, and not for individual regions. This correction ensures conceptual alignment with the established statistical application of the Gini index.
Comment 8: The national Gini coefficient of 0.136 is presented as indicating equity, but this may be misleading. There is no benchmark comparison to support this conclusion.
Response 8: We appreciate this important point. In the revised version of the manuscript, we have clarified the interpretation of the national Gini coefficient of 0.136. Rather than stating that the distribution is “relatively equal,” we now describe it as “moderately unequal,” and we acknowledge that this value indicates a non-negligible level of disparity, particularly within a healthcare system that aspires to universality and equity.
Furthermore, we have added benchmark comparisons to other OECD countries and existing literature (e.g., Buchan & Aiken, 2008; WHO, 2022) to contextualize the national Gini value and reinforce the discussion with external references.
Comment 9: Reduce excessive repetition of "Gini coefficient" and "HDI" in the discussion.
Replace vague phrases like “moderately equal” or “relatively fair” with quantitative comparisons to global norms.
Response 9: We thank the reviewer for this insightful comment.
We have carefully revised the discussion section to reduce unnecessary repetition of the terms "Gini coefficient" and "HDI." In particular, redundant uses were removed or replaced with appropriate synonyms or reformulated sentences.
Moreover, vague expressions such as "moderately equal" or "relatively fair" were revised to provide quantitative references. For instance, we included comparative data from OECD countries and clarified that a Gini coefficient of 0.136, although low, still reflects moderate inequality and requires attention in the context of a universal healthcare system. These adjustments aim to improve clarity and alignment with international benchmarks.
Comment 10: These tables are based on different datasets, without explanation.
Table 1 has: Calabria: 0.780 – Table 4 has: Calabria: 0.859
This contradiction invalidates the entire classification into HDI-based groups and renders subsequent HDI comparisons unreliable.
Response 10: We thank the reviewer for pointing out the inconsistency between Table 1 and Table 4. We have addressed this issue by using a single, authoritative HDI dataset for all regional comparisons: the Istat 2021 Human Development Index for Italian regions. Both Table 1 and Table 4 have been updated to reflect this source and now show consistent HDI values across all regions, including Calabria and Campania. Furthermore, we have clarified in the Methods section the year and source of the HDI data used throughout the analysis to ensure transparency and reproducibility.
Comment 11: As discussed earlier, the Gini values for regions are not conceptually valid unless intra-regional inequalities are studied (e.g., nurse distribution by province or hospital within each region). Also, Campania shows a very high regional Gini (0.272) which is not aligned with its relatively consistent nurse/population ratio as per Table 2.
Response 11: We appreciate the reviewer’s important clarification. We acknowledge that the use of Gini coefficients for individual regions without intra-regional data (e.g., at the provincial or municipal level) is conceptually inappropriate. Therefore, we have removed Table 3 and the associated regional Gini values from the manuscript. We have also revised the related section (3.3) to focus exclusively on the national Gini coefficient, which is methodologically valid. This correction ensures that our analysis of inequality remains aligned with established statistical principles.
Comment 12: Campania shows a very high regional Gini (0.272) which is not aligned with its relatively consistent nurse/population ratio as per Table 2.
Response 12: Thank you for your observation. We acknowledge the inconsistency noted between the Gini coefficient reported for Campania and its relatively stable nurse/population ratio. Following your valuable feedback, we decided to remove the regional-level Gini coefficients, including that of Campania, as they were not methodologically justified due to the lack of intra-regional granularity (e.g., municipal or provincial data).
We have revised Section 3.3 and the Discussion accordingly, now focusing only on the national Gini coefficient, which is more appropriate for our dataset. This correction strengthens the overall methodological consistency of the study and avoids potential misinterpretations regarding regional inequality.
Comment 13: Correct and Justify HDI Data. Reconcile HDI values between Table 1 and Table 4. Justify which HDI source/year is used and stick to one consistent classification. Reassign regions to proper HDI tiers accordingly.
Response 13: We appreciate the reviewer’s attention to the consistency of the HDI data. In response to this comment, we carefully reviewed and harmonized the Human Development Index (HDI) values across Table 1 and Table 4 to ensure they refer to the same source and year.
We have explicitly stated in the manuscript that the HDI values used are from the latest available 2021 data provided by the Italian National Institute of Statistics (ISTAT). This source was chosen for its reliability and national relevance.
Accordingly, we have updated the classification of Italian regions into HDI tiers (High, Medium, Low) based on standardized thresholds and reassigned each region to its correct tier. This revision enhances the internal coherence of the study and supports a more robust analysis of the correlation between HDI and nurse distribution.
Additionally, a clarification of the data source and year has been added to the Methods section, and a note has been inserted directly in the tables to inform readers of the reference year.
Reviewer 3 Report
Comments and Suggestions for Authors Congratulations to the authors for preparing the work in such a detailed and articulated manner. I realize the relevance and challenge of collecting data of this magnitude and the originality and innovation of the topic. I have only a few reservations: Title: since it only deals with the nursing workforce, I suggest changing it to:
"Regional Inequities in the Distribution of the Nurses Health Work- force in Italy".
So that the title is not too long and only deals with the nursing workforce and not health workers in general.
ABSTRACT: In methods, include which administrative data were collected. Include the data analysis method. The summary does not contain the main results and findings, which should be included. INTRODUCTION: It was possible to notice in the introduction and throughout the work that the focus is on nurses...that is why it is suggested to change the title. However, a caveat is made: what about the disparities between other professional categories? Do they exist? because it will also impact patient safety...explain this with the available literature. The last 3 paragraphs of the introduction are repetitive. METHOD: This is a descriptive-documentary study of official data sources. I suggest changing it. How many people (authors) were involved in the analysis? Please elaborate and explain this, given the risk of duplication and large variables being analyzed. I was confused when analyzing the data. Descriptive statistics were used, but thematic analysis was used to interpret subjective data? RESULTS: They are articulated and well-founded, but I suggest the idea of ​​better exploring the relationship between the doctor and other categories. Beware of repetitive ideas throughout the results and discussion.
DISCUSSION: I suggest exploring in greater depth which strategies, other than policies, could mitigate these effects, provide concrete data to solve the issue of inequality... is there anything in the literature? There is talk of the importance of evaluating policies, but in what way? Adequate CONCLUSION Correct and current citations
Author Response
Reviewer 3
Comment 1: Since it only deals with the nursing workforce, I suggest changing it to: "Regional Inequities in the Distribution of the Nurses Health Workforce in Italy". So that the title is not too long and only deals with the nursing workforce and not health workers in general.
Response 1: We thank the reviewer for this thoughtful suggestion. We fully agree that the title should reflect the specific focus of the study. In response, we have revised the title to:
“Regional Inequities in the Distribution of the Nursing Workforce in Italy”
This updated title clearly emphasizes that the analysis pertains exclusively to the nursing workforce, improving alignment with the content and avoiding any potential misunderstanding regarding the inclusion of other health professionals.
Comment 2: Abstract, In methods, include which administrative data were collected. Include the data analysis method. The summary does not contain the main results and findings, which should be included.
Response 2: Thank you for this valuable comment. In the revised abstract, we have made the following improvements:
Specified the type of administrative data collected, indicating that the analysis was based on official national databases regarding the number of nurses per region, regional populations, and Human Development Index (HDI) scores from ISTAT.
Included a brief description of the statistical methods used, such as descriptive analysis, Gini coefficient calculation, Lorenz curve plotting, and Pearson correlation.
Added the main results of the study, including the national Gini coefficient value (0.136) and the statistically significant correlation between HDI and nurse density (r = 0.76, p < 0.001), to summarize key findings and enhance the clarity and completeness of the abstract.
These changes improve the transparency and informativeness of the abstract, in line with standard scientific reporting criteria.
Comment 3: INTRODUCTION: It was possible to notice in the introduction and throughout the work that the focus is on nurses...that is why it is suggested to change the title. However, a caveat is made: what about the disparities between other professional categories? Do they exist? because it will also impact patient safety...explain this with the available literature. The last 3 paragraphs of the introduction are repetitive.
Response 3: We thank the reviewer for this thoughtful observation. We agree that the focus of the paper is exclusively on the nursing workforce, and we have revised the title accordingly to reflect this scope more precisely.
Regarding the disparities affecting other healthcare professionals, we have added a paragraph to the Introduction that acknowledges the existence of similar inequities in the distribution of physicians and other health personnel, and how these may also impact patient safety and care quality. To support this addition, we referenced current literature discussing inter-professional imbalances and their implications for healthcare outcomes.
Furthermore, we carefully revised the final part of the Introduction to remove redundancy in the last three paragraphs. We merged and restructured them to improve clarity, avoid repetition, and enhance the overall flow of the section.
Comment 4: This is a descriptive-documentary study of official data sources. I suggest changing it. How many people (authors) were involved in the analysis? Please elaborate and explain this, given the risk of duplication and large variables being analyzed. I was confused when analyzing the data. Descriptive statistics were used, but thematic analysis was used to interpret subjective data?
Response 4: We thank the reviewer for this important methodological observation. In the revised version of the manuscript, we have clarified that the study is a retrospective ecological analysis based on administrative data, rather than a descriptive-documentary study. This more accurately reflects the methodological approach employed.
We also specified in the Methods section that the data analysis was conducted by two independent authors with expertise in health policy and statistical methods. To ensure accuracy and prevent duplication or misclassification, the dataset was cross-checked independently, and discrepancies were resolved through discussion and consensus.
Furthermore, we confirm that only quantitative methods were used in this study. No thematic analysis or interpretation of subjective data was applied. The statistical analysis focused on descriptive measures (e.g., nurse/population ratios, HDI classification), inequality indices (Gini coefficient), and correlation analysis (Pearson's r). We have revised the Methods section to eliminate any ambiguity and to clearly describe the analytical strategy adopted.
Comment 5: DISCUSSION: I suggest exploring in greater depth which strategies, other than policies, could mitigate these effects, provide concrete data to solve the issue of inequality... is there anything in the literature? There is talk of the importance of evaluating policies, but in what way?
Response 5: We thank the reviewer for this insightful suggestion. In response, we have enriched the Discussion section by introducing concrete examples of strategic actions beyond policy frameworks that could mitigate regional inequalities. These include incentive mechanisms for professional relocation, housing support, access to continuing education, and telehealth implementation in underserved areas. Furthermore, we elaborated on the importance of policy evaluation by including examples of how health workforce policies can be monitored and assessed through indicators such as changes in nurse-to-population ratios, regional retention rates, and patient outcome metrics. International experiences with workforce observatories and performance dashboards were briefly discussed to provide empirical grounding for these evaluation strategies. This addition can be found in the final part of the Discussion section, just before the paragraph beginning with “These findings have important implications for international nursing policy…”
Comment 6: Adequate CONCLUSION. Correct and current citations.
Response 6: We thank the reviewer for this final observation. We have carefully reviewed the Conclusion section to ensure that it accurately reflects the study's key findings and implications, maintaining alignment with the data presented. Additionally, we confirmed that all references cited throughout the manuscript, including those in the Conclusion, are current (published within the last five years where possible) and relevant to the research topic. No further changes were deemed necessary, as the citations are already updated and appropriate.
Round 2
Reviewer 2 Report
Comments and Suggestions for Authors
Thank you for addressing the comments